# Uniform Sampling over Episode Difficulty

**Sébastien M. R. Arnold**[1]*, **Guneet S. Dhillon**[2]*, **Avinash Ravichandran**[3], **Stefano Soatto**[3,4]
[1]University of Southern California, [2]University of Oxford,
[3]Amazon Web Services, [4]University of California, Los Angeles
seb.arnold@usc.edu, guneet.dhillon@stats.ox.ac.uk,
ravinash@amazon.com, soattos@amazon.com

## Abstract

Episodic training is a core ingredient of few-shot learning to train models on tasks with limited labelled data. Despite its success, episodic training remains largely understudied, prompting us to ask the question: what is the best way to sample episodes? In this paper, we first propose a method to approximate episode sampling distributions based on their difficulty. Building on this method, we perform an extensive analysis and find that sampling uniformly over episode difficulty outperforms other sampling schemes, including curriculum and easy-/hard-mining. As the proposed sampling method is algorithm agnostic, we can leverage these insights to improve few-shot learning accuracies across many episodic training algorithms. We demonstrate the efficacy of our method across popular few-shot learning datasets, algorithms, network architectures, and protocols.

## 1 Introduction

Large amounts of high-quality data have been the key for the success of deep learning algorithms. Furthermore, factors such as data augmentation and sampling affect model performance significantly. Continuously collecting and curating data is a resource (cost, time, storage, etc.) intensive process. Hence, recently, the machine learning community has been exploring methods for performing transfer-learning from large datasets to unseen tasks with limited data.

A popular genre of these approaches is called meta-learning few-shot approaches, where, in addition to the limited data from the task of interest, a large dataset of disjoint tasks is available for (pre-)training. These approaches are prevalent in the area of computer vision [31] and reinforcement learning [6]. A key component of these methods is the notion of episodic training, which refers to sampling tasks from the larger dataset for training. By learning to solve these tasks correctly, the model can generalize to new tasks.

However, sampling for episodic training remains surprisingly understudied despite numerous methods and applications that build on it. To the best of our knowledge, only a handful of works [69, 55, 33] explicitly considered the consequences of sampling episodes. In comparison, stochastic [46] and mini-batch [4] sampling alternatives have been thoroughly analyzed from the perspectives of optimization [17, 5], information theory [27, 9], and stochastic processes [68, 66], among many others. Building a similar understanding of sampling for episodic training will help theoreticians and practitioners develop improved sampling schemes, and is thus of crucial importance to both.

In this paper, we explore many sampling schemes to understand their impact on few-shot methods. Our work revolves around the following fundamental question: *what is the best way to sample episodes?* Our focus will be restricted to image classification in the few-shot learning setting – where "best" is taken to mean "higher transfer accuracy of unseen episodes" – and leave analyses and applications to other areas for future work.

---

*Equal contributions; work done while at Amazon Web Services.

35th Conference on Neural Information Processing Systems (NeurIPS 2021).

Contrary to prior work, our experiments indicate that sampling uniformly with respect to *episode difficulty* yields higher classification accuracy – a scheme originally proposed to regularize metric learning [62]. To better understand these results, we take a closer look at the properties of episodes and what makes them difficult. Building on this understanding, we propose a method to approximate different sampling schemes, and demonstrate its efficacy on several standard few-shot learning algorithms and datasets.

Concretely, we make the following contributions:

- We provide a detailed empirical analysis of episodes and their difficulty. When sampled randomly, we show that episode difficulty (approximately) follows a normal distribution and that the difficulty of an episode is largely independent of several modeling choices including the training algorithm, the network architecture, and the training iteration.
- Leveraging our analysis, we propose *simple and universally applicable* modifications to the episodic sampling pipeline to approximate *any* sampling scheme. We then use this scheme to thoroughly compare episode sampling schemes – including easy/hard-mining, curriculum learning, and uniform sampling – and report that sampling uniformly over episode difficulty yields the best results.
- Finally, we show that *sampling matters for few-shot classification* as it improves transfer accuracy for a diverse set of popular [53, 20, 15, 41] and state-of-the-art [64] algorithms on standard and cross-domain benchmarks.

## 2 Preliminaries

### 2.1 Episodic sampling and training

We define episodic sampling as subsampling few-shot tasks (or episodes) from a larger *base* dataset [7]. Assuming the base dataset admits a generative distribution, we sample an episode in two steps[2]. First, we sample the episode classes $\mathcal{C}_\tau$ from class distribution $p(\mathcal{C}_\tau)$; second, we sample the episode's data from data distribution $p(x, y \mid \mathcal{C}_\tau)$ conditioned on $\mathcal{C}_\tau$. This gives rise to the following log-likelihood for a model $l_\theta$ parameterized by $\theta$:

$$\mathcal{L}(\theta) = \operatorname*{E}_{\tau \sim q(\cdot)} \left[ \log l_\theta(\tau) \right], \tag{1}$$

where $q(\tau)$ is the episode distribution induced by first sampling classes, then data. In practice, this expectation is approximated by sampling a batch of episodes $\mathcal{B}$ each with their set $\tau_Q$ of *query* samples. To enable transfer to unseen classes, it is also common to include a small set $\tau_S$ of *support* samples to provide statistics about $\tau$. This results in the following Monte-Carlo estimator:

$$\mathcal{L}(\theta) \approx \frac{1}{|\mathcal{B}|} \sum_{\tau \in \mathcal{B}} \frac{1}{|\tau_Q|} \sum_{(x,y) \in \tau_Q} \log l_\theta(y \mid x, \tau_S), \tag{2}$$

where the data in $\tau_Q$ and $\tau_S$ are both distributed according to $p(x, y \mid \mathcal{C}_\tau)$. In few-shot classification, the $n$-way $k$-shot setting corresponds to sampling $n = |\mathcal{C}_\tau|$ classes, each with $k = \frac{|\tau_S|}{n}$ support data points. The implicit assumption in the vast majority of few-shot methods is that both classes and data are sampled with *uniform* probability – *but are there better alternatives?* We carefully examine these underlying assumptions in the forthcoming sections.

### 2.2 Few-shot algorithms

We briefly present a few representative episodic learning algorithms. A more comprehensive treatment of few-shot algorithms is presented in Wang et al. [59] and Hospedales et al. [24]. A core question in few-shot learning lies in evaluating (and maximizing) the model likelihood $l_\theta$. These algorithms can be divided in two major families: gradient-based methods, which adapt the model's parameters to the episode; and metric-based methods, which compute similarities between support and query samples in a learned embedding space.

Gradient-based few-shot methods are best illustrated through Model-Agnostic Meta-Learning [15] (MAML). The intuition behind MAML is to learn a set of initial parameters which can quickly

---

[2]We use the notation for a probability distribution and its probability density function interchangeably.

specialize to the task at hand. To that end, MAML computes $l_\theta$ by adapting the model parameters $\theta$ via one (or more) steps of gradient ascent and then computes the likelihood using the adapted parameters $\theta'$. Concretely, we first compute the likelihood $p_\theta(y \mid x)$ using the support set $\tau_S$, adapt the model parameters, and then evaluate the likelihood:

$$l_\theta(y \mid x, \tau_S) = p_{\theta'}(y \mid x) \quad \text{s.t.} \quad \theta' = \theta + \alpha \nabla_\theta \sum_{(x,y) \in \tau_S} \log p_\theta(y \mid x),$$

where $\alpha > 0$ is known as the adaptation learning rate. A major drawback of training with MAML lies in back-propagating through the adaptation phase, which requires higher-order gradients. To alleviate this computational burden, Almost No Inner Loop [41] (ANIL) proposes to only adapt the last classification layer of the model architecture while tying the rest of the layers across episodes. They empirically demonstrate little classification accuracy drop while accelerating training times four-fold.

Akin to ANIL, metric-based methods also share most of their parameters across tasks; however, their aim is to learn a metric space where classes naturally cluster. To that end, metric-based algorithms learn a feature extractor $\phi_\theta$ parameterized by $\theta$ and classify according to a non-parametric rule. A representative of this family is Prototypical Network [53] (ProtoNet), which classifies query points according to their distance to *class prototypes* – the average embedding of a class in the support set:

$$l_\theta(y \mid x, \tau_S) = \frac{\exp\left(-d(\phi_\theta(x), \phi_\theta^y)\right)}{\sum_{y' \in \mathcal{C}_\tau} \exp\left(-d(\phi_\theta(x), \phi_\theta^{y'})\right)} \quad \text{s.t.} \quad \phi_\theta^c = \frac{1}{k} \sum_{\substack{(x,y) \in \tau_S \\ y=c}} \phi_\theta(x),$$

where $d(\cdot, \cdot)$ is a distance function such as the Euclidean distance or the negative cosine similarity, and $\phi_\theta^c$ is the class prototype for class $c$. Other classification rules include support vector clustering [32], neighborhood component analysis [30], and the earth-mover distance [67].

### 2.3 Episode difficulty

Given an episode $\tau$ and likelihood function $l_\theta$, we define *episode difficulty* to be the negative log-likelihood incurred on that episode:

$$\Omega_{l_\theta}(\tau) = -\log l_\theta(\tau),$$

which is a surrogate for how hard it is to classify the samples in $\tau_Q$ correctly, given $l_\theta$ and $\tau_S$. By definition, this choice of episode difficulty is tied to the choice of the likelihood function $l_\theta$.

Dhillon et al. [11] use a similar surrogate as a means to systematically report few-shot performances. We use this definition because it is equivalent to the loss associated with the likelihood function $l_\theta$ on episode $\tau$, which is readily available at training time.

## 3 Methodology

In this section, we describe the core assumptions and methodology used in our study of sampling methods for episodic training. Our proposed method builds on importance sampling [21] (IS) which we found compelling for three reasons: (i) IS is *well understood* and solidly grounded from a theoretical standpoint, (ii) IS is *universally applicable* thus compatible with all episodic training algorithms, and (iii) IS is *simple to implement* with little requirement for hyper-parameter tuning.

*Why should we care about episodic sampling?* A back-of-the-envelope calculation[3] suggests that there are on the order of $10^{162}$ different training episodes for the smallest-scale experiments in Section 5. Since iterating through each of them is infeasible, we ought to express some preference over which episodes to sample. In the following, we describe a method that allows us to specify this preference.

### 3.1 Importance sampling for episodic training

Let us assume that the sampling scheme described in Section 2.1 induces a distribution $q(\tau)$ over episodes. We call it the *proposal distribution*, and assume knowledge of its density function. We wish

---

[3]For a base dataset with $N$ classes and $K$ input-output pairs per class, there are a total of $\binom{N}{n}\binom{K}{k}^n$ possible episodes that can be created when sampling $k$ pairs each from $n$ classes.

---

**Algorithm 1:** Episodic training with Importance Sampling

---

**Input:** target ($p$) and proposal ($q$) distributions, likelihood function $l_\theta$, optimizer OPT.
Randomly initialize model parameters $\theta$.
**repeat**
    Sample a mini-batch $\mathcal{B}$ of episodes from $q(\tau)$.
    **for** each episode $\tau$ in mini-batch $\mathcal{B}$ **do**
        Compute episode likelihood: $l_\theta(\tau)$.
        Compute importance weight: $w(\tau) = \frac{p(\tau)}{q(\tau)}$.
    **end for**
    Aggregate: $\mathcal{L}(\theta) \leftarrow \sum_{\tau \in \mathcal{B}} w(\tau) \log l_\theta(\tau)$.
    Compute effective sample size $\text{ESS}(\mathcal{B})$.
    Update model parameters: $\theta \leftarrow \text{OPT}(\frac{\mathcal{L}(\theta)}{\text{ESS}(\mathcal{B})})$.
**until** parameters $\theta$ have converged.

---

to estimate the expectation in Eq. (1) when sampling episodes according to a *target distribution* $p(\tau)$ of our choice, rather than $q(\tau)$. To that end, we can use an importance sampling estimator which simply re-weights the observed values for a given episode $\tau$ by $w(\tau) = \frac{p(\tau)}{q(\tau)}$, the ratio of the target and proposal distributions:

$$\mathop{\mathrm{E}}_{\tau \sim p(\cdot)} [\log l_\theta(\tau)] = \mathop{\mathrm{E}}_{\tau \sim q(\cdot)} [w(\tau) \log l_\theta(\tau)].$$

The importance sampling identity holds whenever $q(\tau)$ has non-zero density over the support of $p(\tau)$, and effectively allows us to sample from *any* target distribution $p(\tau)$.

A practical issue of the IS estimator arises when some values of $w(\tau)$ become much larger than others; in that case, the likelihoods $l_\theta(\tau)$ associated with mini-batches containing heavier weights dominate the others, leading to disparities. To account for this effect, we can replace the mini-batch average in the Monte-Carlo estimate of Eq. (2) by the *effective sample size* $\text{ESS}(\mathcal{B})$ [29, 34]:

$$\mathop{\mathrm{E}}_{\tau \sim p(\cdot)} [\log l_\theta(\tau)] \approx \frac{1}{\text{ESS}(\mathcal{B})} \sum_{\tau \in \mathcal{B}} w(\tau) \log l_\theta(\tau) \quad \text{s.t.} \quad \text{ESS}(\mathcal{B}) = \frac{(\sum_{\tau \in \mathcal{B}} w(\tau))^2}{\sum_{\tau \in \mathcal{B}} w(\tau)^2}, \qquad (3)$$

where $\mathcal{B}$ denotes a mini-batch of episodes sampled according to $q(\tau)$. Note that when $w(\tau)$ is constant, we recover the standard mini-batch average setting as $\text{ESS}(\mathcal{B}) = |\mathcal{B}|$. Empirically, we observed that normalizing with the effective sample size avoided instabilities. This method is summarized in Algorithm 1.

### 3.2 Modeling the proposal distribution

A priori, we do not have access to the proposal distribution $q(\tau)$ (nor its density) and thus need to estimate it empirically. Our main assumption is that sampling episodes from $q(\tau)$ induces a normal distribution over episode difficulty. With this assumption, we model the proposal distribution by this induced distribution, therefore replacing $q(\tau)$ with $\mathcal{N}(\Omega_{l_\theta}(\tau) \mid \mu, \sigma^2)$ where $\mu, \sigma^2$ are the mean and variance parameters. As we will see in Section 5.2, this normality assumption is experimentally supported on various datasets, algorithms, and architectures.

We consider two settings for the estimation of $\mu$ and $\sigma^2$: offline and online. The *offline* setting consists of sampling $1,000$ training episodes before training, and computing $\mu, \sigma^2$ using a model pre-trained on the same base dataset. Though this setting seems unrealistic, *i.e.* having access to a pre-trained model, several meta-learning few-shot methods start with a pre-trained model which they further build upon. Hence, for such methods there is no overhead. For the *online* setting, we estimate the parameters on-the-fly using the model currently being trained. This is justified by the analysis in Section 5.2 which shows that episode difficulty transfers across model parameters during training. We update our estimates of $\mu, \sigma^2$ with an exponential moving average:

$$\mu \leftarrow \lambda\mu + (1 - \lambda)\Omega_{l_\theta}(\tau) \quad \text{and} \quad \sigma^2 \leftarrow \lambda\sigma^2 + (1 - \lambda)(\Omega_{l_\theta}(\tau) - \mu)^2,$$

where $\lambda \in [0, 1]$ controls the adjustment rate of the estimates, and the initial values of $\mu, \sigma^2$ are computed in a warm-up phase lasting 100 iterations. Keeping $\lambda = 0.9$ worked well for all our

experiments (Section 5). We opted for this simple implementation as more sophisticated approaches like West [61] yielded little to no benefit.

### 3.3 Modeling the target distribution

Similar to the proposal distribution, we model the target distribution by its induced distribution over episode difficulty. Our experiments compare four different approaches, all of which share parameters $\mu, \sigma^2$ with the normal model of the proposal distribution. For numerical stability, we truncate the support of all distributions to $[\mu - 2.58\sigma, \mu + 2.58\sigma]$, which gives approximately $99\%$ coverage for the normal distribution centered around $\mu$.

The first approach (HARD) takes inspiration from hard negative mining [51], where we wish to sample only more challenging episodes. The second approach (EASY) takes a similar view but instead only samples easier episodes. We can model both distributions as follows:

$$\mathcal{U}(\Omega_{l_\theta}(\tau) \mid \mu, \mu + 2.58\sigma) \qquad \text{(HARD)}$$
$$\text{and}$$
$$\mathcal{U}(\Omega_{l_\theta}(\tau) \mid \mu - 2.58\sigma, \mu) \qquad \text{(EASY)}$$

where $\mathcal{U}$ denotes the uniform distribution. The third (CURRICULUM) is motivated by curriculum learning [2], which slowly increases the likelihood of sampling more difficult episodes:

$$\mathcal{N}(\Omega_{l_\theta}(\tau) \mid \mu_t, \sigma^2) \qquad \text{(CURRICULUM)}$$

where $\mu_t$ is linearly interpolated from $\mu - 2.58\sigma$ to $\mu + 2.58\sigma$ as training progresses. Finally, our fourth approach, UNIFORM, resembles distance weighted sampling [62] and consists of sampling uniformly over episode difficulty:

$$\mathcal{U}(\Omega_{l_\theta}(\tau) \mid \mu - 2.58\sigma, \mu + 2.58\sigma). \qquad \text{(UNIFORM)}$$

Intuitively, UNIFORM can be understood as a uniform prior over unseen test episodes, with the intention of performing well across the entire difficulty spectrum. This acts as a regularizer, forcing the model to be equally discriminative for both easy and hard episodes.

## 4 Related Works

This paper studies task sampling in the context of few-shot [36, 14] and meta-learning [49, 56].

**Few-shot learning.** This setting has received a lot of attention over recent years [58, 43, 47, 18]. Broadly speaking, state-of-the-art methods can be categorized in two major families: metric-based and gradient-based.

Metric-based methods learn a shared feature extractor which is used to compute the distance between samples in embedding space [53, 3, 44, 30]. The choice of metric mostly differentiates one method from another; for example, popular choices include Euclidean distance [53], negative cosine similarity [20], support vector machines [32], set-to-set functions [64], or the earth-mover distance [67].

Gradient-based algorithms such as MAML [15], propose an objective to learn a network initialization that can quickly adapt to new tasks. Due to its minimal assumptions, MAML has been extended to probabilistic formulations [22, 65] to incorporate learned optimizers – implicit [16] or explicit [40] – and simplified to avoid expensive second-order computations [37, 42]. In that line of work, ANIL [41] claims to match MAML's performance when adapting only the last classification layer – thus greatly reducing the computational burden and bringing gradient and metric-based methods closer together.

**Sampling strategies.** Sampling strategies have been studied for different training regimes. Wu et al. [62] demonstrate that "sampling matters" in the context of metric learning. They propose to sample a triplet with probability proportional to the distance of its positive and negative samples, and observe stabilized training and improved accuracy. This observation was echoed by Katharopoulos and Fleuret [27] when sampling mini-batches: carefully choosing the constituent samples of a mini-batch improves the convergence rate and asymptotic performance. Like ours, their method builds on importance sampling [52, 12, 26] but whereas they compute importance weights using the magnitude

of the model's gradients, we use the episode's difficulty. Similar insights were also observed in reinforcement learning, where Schaul et al. [48] suggests a scheme to sample transitions according to the temporal difference error.

Closer to our work, Sun et al. [55] present a hard-mining scheme where the most challenging classes across episodes are pooled together and used to create new episodes. Observing that the difficulty of a class is intrinsically linked to the other classes in the episode, Liu et al. [33] propose a mechanism to track the difficulty across every class pair. They use this mechanism to build a curriculum [2, 63] of increasingly difficult episodes. In contrast to these two approaches, our proposed method makes use of importance sampling to mimic the target distribution rather than sampling from it directly. This helps achieve fast and efficient sampling without any preprocessing requirements.

## 5 Experiments

We first validate the assumptions underlying our proposed IS estimator and shed light on the properties of episode difficulty. Then, we answer the question we pose in the introduction, namely: *what is the best way to sample episodes?* Finally, we ask if better sampling improves few-shot classification.

### 5.1 Experimental setup

We review the standardized few-shot benchmarks and provide a detailed description in Appendix A.

**Datasets.** We use two standardized image classification datasets, Mini-ImageNet [58] and Tiered-ImageNet [45], both subsets of ImageNet [10]. Mini-ImageNet consists of $64$ classes for training, $16$ for validation, and $20$ for testing; we use the class splits introduced by Ravi and Larochelle [43]. Tiered-ImageNet contains $608$ classes split into $351$, $97$, and $160$ for training, validation, and testing, respectively.

**Network architectures.** We train two model architectures. A 4-layer convolution network conv$(64)_4$ Vinyals et al. [58] with $64$ channels per layer. And ResNet-12, a 12-layer deep residual network [23] introduced by Oreshkin et al. [39]. Both architectures use batch normalization [25] after every convolutional layer and ReLU as the non-linearity.

**Training algorithms.** For the metric-based family, we use ProtoNet with Euclidean [53] and scaled negative cosine similarity measures [20]. Additionally, we use MAML [15] and ANIL [41] as representative gradient-based algorithms.

**Hyper-parameters.** We tune hyper-parameters for each algorithm and dataset to work well across different few-shot settings and network architectures. Additionally, we keep the hyper-parameters the same across all different sampling methods for a fair comparison. We train for 20k iterations with a mini-batch of size 16 and 32 for Mini-ImageNet and Tiered-ImageNet respectively, and validate every 1k iterations on 1k episodes. The best performing model is finally evaluated on 1k test episodes.

### 5.2 Understanding episode difficulty

All the models in this subsection are trained using baseline sampling as described in Section 2.1, *i.e.*, episodic training without importance sampling.

#### 5.2.1 Episode difficulty is approximately normally distributed

We begin our analysis by verifying that the distribution over episode difficulty induced by $q(\tau)$ is approximately normal. In Fig. 1, we use the difficulty of 10k test episodes sampled with $q(\tau)$. The difficulties are computed using conv$(64)_4$ trained with ProtoNet and MAML on Mini-ImageNet for 1-shot 5-way classification. The episode difficulty density plots follow a bell curve, which are naturally modeled with a normal distribution. The Q-Q plots, typically used to assess normality, suggest the same – the closer the curve is to the identity line, the closer the distribution is to a normal.

Finally, we compute the Shapiro-Wilk test for normality [50], which tests for the null hypothesis that the data is drawn from a normal distribution. Since the p-value for this test is sensitive to the

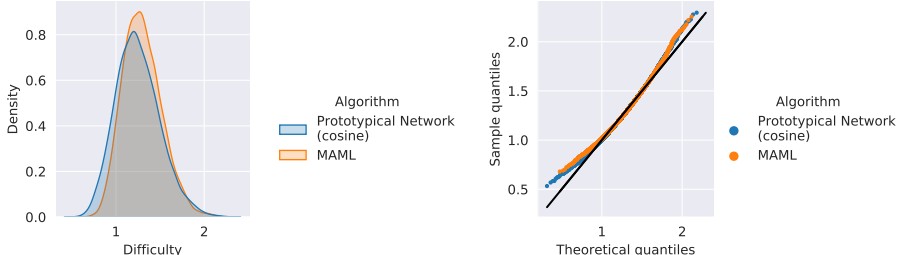

Figure 1: **Episode difficulty is approximately normally distributed.** Density (left) and Q-Q (right) plots of the episode difficulty computed by conv$(64)_4$'s on Mini-ImageNet (1-shot 5-way), trained using ProtoNets (cosine) and MAML (depicted in the legends). The values are computed over 10k test episodes. The density plots follow a bell curve, with the density peak in the middle, which quickly drops-off on either side of the peak. The Q-Q plots are close to the identity line (in black). The closer the curve is to the identity line, the closer the distribution is to a normal. Both suggest that the episode difficulty distribution can be normally approximated.

sample size[4], we subsample 50 values 100 times and average rejection rates over these subsets. With $\alpha = 0.05$, the null hypothesis is rejected 14% and 17% of the time for Mini-ImageNet and Tiered-ImageNet respectively, thus suggesting that episode difficulty can be reliably approximated with a normal distribution.

### 5.2.2 Independence from modeling choices

By definition, the notion of episode difficulty is tightly coupled to the model likelihood $l_\theta$ (Section 2.3), and hence to the modeling variables such as learning algorithm, network architecture, and model parameters. We check if episode difficulty transfers across different choices for these variables.

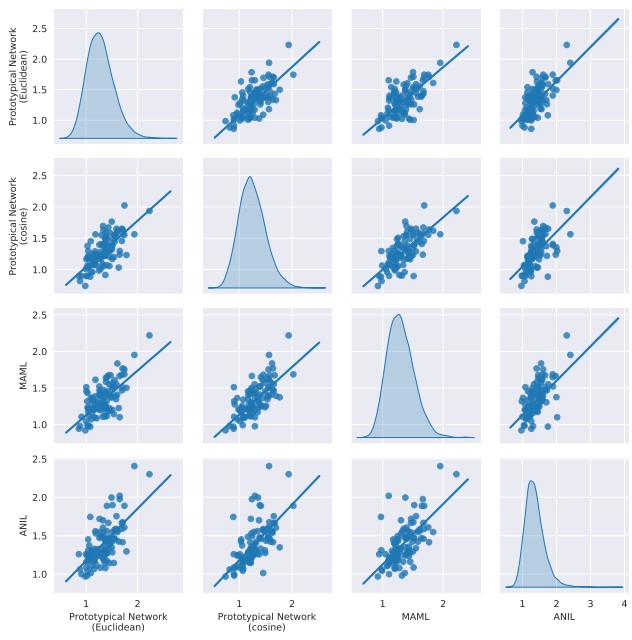

We are concerned with the *relative ranking* of the episode difficulty and not the actual values. To this end, we will use the Spearman rank-order correlation coefficient, a non-parametric measure of the monotonicity of the relationship between two sets of values. This value lies within $[-1; 1]$, with $0$ implying no correlation, and $+1$ and $-1$ implying exact positive and negative correlations, respectively.

**Training algorithm.** We first check the dependence on the training algorithm. We use all four algorithms to train conv$(64)_4$'s for 1-shot 5-way classification on Mini-ImageNet, then compute episode difficulty over 10k test episodes. The Spearman rank-order correlation coefficients for the difficulty values computed with respect to all possible pairs of training algorithms are $> 0.65$. This positive correlation is illustrated in Fig. 2 and suggests that an episode that is difficult for one training algorithm is very likely to be difficult for another.

Figure 2: **Episode difficulty transfers across training algorithms.** Scatter plots (with regression lines) of the episode difficulty computed on 1k Mini-ImageNet test episodes (1-shot 5-way) by conv$(64)_4$'s trained using different algorithms. The positive correlation suggests that an episode that is difficult for one training algorithm will be difficult for another.

**Network architecture.** Next we analyze the dependence on the network architecture. We trained conv$(64)_4$ and ResNet-12's using all training algorithms for Mini-ImageNet 1-shot 5-way classi-

---

[4]For a large sample size, the p-values are not reliable as they may detect trivial departures from normality.

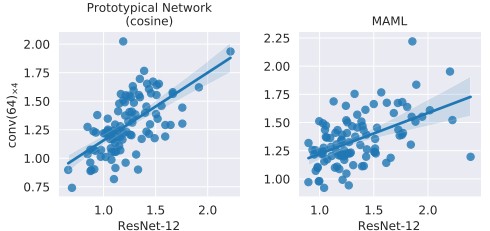 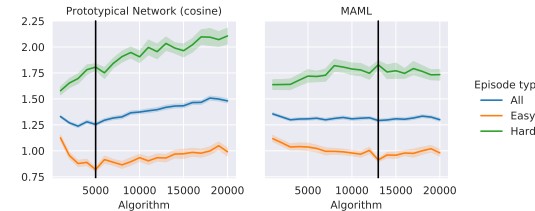

Figure 3: **Episode difficulty transfers across network architectures.** Scatter-plots (with regression lines) of the episode difficulty computed by conv$(64)_4$ and ResNet-12's trained using different algorithms. This is computed for 10k 1-shot 5-way test episodes from Mini-ImageNet. We observe a strong positive correlation between the computed values for both network architectures.

Figure 4: **Episode difficulty is transferred across model parameters during training.** We select the 50 easiest and hardest episodes and track their difficulty during training. This is done for conv$(64)_4$'s trained on Mini-ImageNet (1-shot 5-way) with different algorithms. The average difficulty of the episodes decreases over time, until convergence (vertical line), after which the model overfits. Additionally, easier episodes remain easy while harder episodes remain hard, indicating that episode difficulty transfers from one set of parameters to the next.

fication. We compute the episode difficulties for 10k test episodes and compute their Spearman rank-order correlation coefficients across the two architectures, for a given algorithm. The correlation coefficients are 0.57 for ProtoNet (Euclidean), 0.72 for ProtoNet (cosine), 0.58 for MAML, and 0.49 for ANIL. Fig. 3 illustrates this positive correlation, suggesting that episode difficulty is transferred across network architectures with high probability.

**Model parameters during training.** Lastly, we study the dependence on model parameters during training. We select the 50 *easiest* and 50 *hardest* episodes, *i.e.*, episodes with the lowest and highest difficulties respectively, from 1k test episodes. We track the episode difficulty for all episodes over the training phase and visualize the trend in Fig. 4 for conv$(64)_4$'s trained using different training algorithms on Mini-ImageNet (1-shot 5-way). Throughout training, easy episodes remain easy and hard episodes remain hard, hence suggesting that episode difficulty transfers across different model parameters during training. Since the episode difficulty does not change drastically during the training process, we can estimate it with a running average over the training iterations. This justifies the *online* modeling of the proposal distribution in Section 3.2.

Table 1: **Few-shot accuracies on benchmark datasets for 5-way few-shot episodes in the offline setting.** Mean accuracy and $95\%$ confidence interval computed over 1k test episodes. The first row in every scenario denotes baseline sampling. Best results for a fixed scenario are shown in bold and the † indicates matching or improving over baseline sampling.

| | Mini-ImageNet | | | | Tiered-ImageNet | |
|---|---|---|---|---|---|---|
| | conv$(64)_4$ | | ResNet-12 | | ResNet-12 | |
| | 1-shot (%) | 5-shot (%) | 1-shot (%) | 5-shot (%) | 1-shot (%) | 5-shot (%) |
| ProtoNet (cosine) | **50.03±0.61** | 61.56±0.53 | 52.85±0.64 | 62.11±0.52 | **60.01±0.73** | 72.75±0.59 |
| + EASY | 49.60±0.61† | 65.17±0.53† | 53.35±0.63† | 63.55±0.53† | **60.03±0.75**† | 74.65±0.57† |
| + HARD | 49.01±0.60 | **66.45±0.50**† | 52.65±0.63† | 70.15±0.51† | 55.44±0.72 | 75.97±0.55† |
| + CURRICULUM | 49.38±0.61 | 64.12±0.53† | 53.21±0.65† | 65.89±0.52† | **60.37±0.76**† | 75.32±0.58† |
| + UNIFORM | **50.07±0.59**† | 66.33±0.52† | **54.27±0.65**† | **70.85±0.51**† | 60.27±0.75† | **78.36±0.54**† |

### 5.3 Comparing episode sampling methods

We compare different methods for episode sampling. To ensure fair comparisons, we use the offline formulation (Section 3.2) so that all sampling methods share the same pre-trained network (the network trained using baseline sampling) when computing proposal likelihoods. We compute results over 2 datasets, 2 network architectures, 4 algorithms and 2 few-shot protocols, totaling in 24 scenarios. Table 1 presents results on ProtoNet (cosine), while the rest are in Appendix D.

We observe that, although not strictly dominant, UNIFORM tends to outperform other methods as it is within the statistical confidence of the best method in 19/24 scenarios. For the 5/24 scenarios

Table 2: **Few-shot accuracies on benchmark datasets for 5-way few-shot episodes in the offline and online settings.** The mean accuracy and the 95% confidence interval are reported for evaluation done over 1k test episodes. The first row in every scenario denotes baseline sampling. Best results for a fixed scenario are shown in bold. Results where a sampling technique is better than or comparable to baseline sampling are denoted by †. UNIFORM (Online) retains most of the performance of the offline formulation while being significantly easier to implement (online is competitive in 15/24 scenarios vs 16/24 for offline).

| | Mini-ImageNet | | | | Tiered-ImageNet | |
| --- | --- | --- | --- | --- | --- | --- |
| | conv(64)$_4$ | | ResNet-12 | | ResNet-12 | |
| | 1-shot (%) | 5-shot (%) | 1-shot (%) | 5-shot (%) | 1-shot (%) | 5-shot (%) |
| ProtoNet (Euclidean) | **49.06±0.60** | 65.28±0.52 | 49.67±0.64 | 67.45±0.51 | **59.10±0.73** | 76.95±0.56 |
| + UNIFORM (Offline) | 48.19±0.62 | 66.73±0.52$^†$ | **53.94±0.63$^†$** | **70.79±0.49$^†$** | 58.63±0.76$^†$ | **78.62±0.55$^†$** |
| + UNIFORM (Online) | 48.39±0.62 | **67.86±0.50$^†$** | 52.97±0.64$^†$ | 70.63±0.50$^†$ | 59.67±0.70$^†$ | 78.73±0.55$^†$ |
| ProtoNet (cosine) | **50.03±0.61** | 61.56±0.53 | 52.85±0.64 | 62.11±0.52 | 60.01±0.73 | 72.75±0.59 |
| + UNIFORM (Offline) | **50.07±0.59$^†$** | 66.33±0.52$^†$ | **54.27±0.65$^†$** | **70.85±0.51$^†$** | 60.27±0.75$^†$ | **78.36±0.54$^†$** |
| + UNIFORM (Online) | 50.06±0.61$^†$ | 65.99±0.52$^†$ | 53.90±0.63$^†$ | 68.78±0.51$^†$ | **61.37±0.72$^†$** | 77.81±0.56$^†$ |
| MAML | **46.88±0.60** | 55.16±0.55 | 49.92±0.65 | 63.93±0.59 | 55.37±0.74 | **72.93±0.60** |
| + UNIFORM (Offline) | 46.67±0.63$^†$ | **62.09±0.55$^†$** | **52.65±0.65$^†$** | **66.76±0.57$^†$** | 54.58±0.77 | 72.00±0.66 |
| + UNIFORM (Online) | 46.70±0.61$^†$ | 61.62±0.54$^†$ | 51.17±0.68$^†$ | 65.63±0.57$^†$ | **57.15±0.74$^†$** | 71.67±0.67 |
| ANIL | **46.59±0.60** | 63.47±0.55 | 49.65±0.65 | 59.51±0.56 | **54.77±0.76** | 69.28±0.67 |
| + UNIFORM (Offline) | **46.93±0.62$^†$** | 62.75±0.60 | 49.56±0.62$^†$ | 64.72±0.60$^†$ | 54.15±0.79$^†$ | **70.44±0.69$^†$** |
| + UNIFORM (Online) | 46.82±0.63$^†$ | 62.63±0.59 | **49.82±0.68$^†$** | 64.51±0.62$^†$ | 55.18±0.74$^†$ | 69.55±0.71$^†$ |

where UNIFORM underperforms, it closely trails behind the best methods. Compared to baseline sampling, the average degradation of UNIFORM is $-0.83\%$ and at most $-1.44\%$ (ignoring the standard deviations) in $4/24$ scenarios. Conversely, UNIFORM boosts accuracy by as much as $8.74\%$ and on average by $3.86\%$ (ignoring the standard deviations) in $13/24$ scenarios. We attribute this overall good performance to the fact that uniform sampling puts a uniform distribution prior over the (unseen) test episodes, with the intention of performing well across the entire difficulty spectrum. This acts as a regularizer, forcing the model to be equally discriminative for easy and hard episodes. If we knew the test episode distribution, upweighting episodes that are most likely under that distribution will improve transfer accuracy [13]. However, this uninformative prior is the safest choice without additional information about the test episodes.

Second best is baseline sampling as it is statistically competitive on $10/24$ scenarios, while EASY, HARD, and CURRICULUM only appear among the better methods in 4, 4, and 9 scenarios, respectively.

### 5.4 Online approximation of the proposal distribution

Although the offline formulation is better suited for analysis experiments, it is expensive as it requires a pre-training phase for the proposal network and 2 forward passes during episodic training (one for the episode loss, another for the proposal density). In this subsection, we show that the online formulation faithfully approximates offline sampling and can retain most of the performance improvements from UNIFORM. We take the same 24 scenarios as in the previous subsection, and compare baseline sampling against offline and online UNIFORM. Table 2 reports the full suite of results.

We observe that baseline is statistically competitive on $9/24$ scenarios; on the other hand, offline and online UNIFORM perform similarly on aggregate, as they are within the best results in $16/24$ and $15/24$ scenarios respectively. Similar to its offline counterpart, online UNIFORM does better than or comparable to baseline sampling in 21 out of 24 scenarios. On the $3/24$ scenarios where online UNIFORM underperforms compared to baseline, the average degradation is $-0.92\%$, and at most $-1.26\%$ (ignoring the standard deviations). Conversely, in the remaining scenarios, it boosts accuracy by as much as $6.67\%$ and on average by $2.24\%$ (ignoring the standard deviations). Therefore, using online UNIFORM, while computationally comparable to baseline sampling, results in a boost in few-shot performance; when it underperforms, it trails closely. We also compute the mean accuracy difference between the offline and online formulation, which is $0.07\% \pm 0.35$ accuracy points. This confirms that both the offline and online methods produce quantitatively similar outcomes.

Table 3: **Few-shot accuracies for 5-way cross-domain few-shot episodes after training on Mini-ImageNet.** The mean accuracy and the $95\%$ confidence interval are reported for evaluation done over 1k test episodes. The first row in every scenario denotes baseline sampling. Best results for a fixed scenario are shown in bold.

|  | CUB-200 | | Describable Textures | |
|---|---|---|---|---|
|  | ResNet-12 | | conv$(64)_4$ | |
|  | 1-shot (%) | 5-shot (%) | 1-shot (%) | 5-shot (%) |
| ProtoNet (cosine) | 38.67±0.60 | 49.75±0.57 | 32.09±0.45 | 38.44±0.41 |
| + UNIFORM (Online) | **40.55±0.60** | **56.30±0.55** | **33.63±0.47** | **43.28±0.44** |
| MAML | 35.80±0.56 | 45.16±0.62 | 29.47±0.46 | 37.85±0.47 |
| + UNIFORM (Online) | **37.18±0.55** | **46.58±0.58** | **31.84±0.49** | **40.81±0.44** |

### 5.5 Better sampling improves cross-domain transfer

To further validate the role of episode sampling as a way to improve generalization, we evaluate the models trained in the previous subsection on episodes from completely different domains. Specifically, we use the models trained on Mini-ImageNet with baseline and online UNIFORM sampling to evaluate on the test episodes of CUB-200 [60], Describable Textures [8], FGVC Aircrafts [35], and VGG Flowers [38], following the splits of Triantafillou et al. [57]. Table 3 displays results for ProtoNet (cosine) and MAML on CUB-200 and Describable Textures, with the complete set of experiments available in Appendix F. Out of the 64 total cross-domain scenarios, online UNIFORM does statistically better in $49/64$ scenarios, comparable in $12/64$ scenarios and worse in only $3/64$ scenarios. These results further go to show that sampling matters in episodic training.

Table 4: **Few-shot accuracies on benchmark datasets for 5-way few-shot episodes using FEAT.** The mean accuracy and the $95\%$ confidence interval are reported for evaluation done over 10k test episodes with a ResNet-12. The first row in every scenario denotes baseline sampling. Best results for a fixed scenario are shown in bold. UNIFORM (Online) improves FEAT's accuracy in $3/4$ scenarios, demonstrating that sampling matters even for state-of-the-art few-shot methods.

|  | Mini-ImageNet | | Tiered-ImageNet | |
|---|---|---|---|---|
|  | 1-shot (%) | 5-shot (%) | 1-shot (%) | 5-shot (%) |
| FEAT | 66.02±0.20 | 81.17±0.14 | **70.50±0.23** | 84.26±0.16 |
| + UNIFORM (Online) | **66.27±0.20** | **81.54±0.14** | **70.61±0.23** | **84.42±0.16** |

### 5.6 Better sampling improves few-shot classification

The results in the previous subsections suggest that online UNIFORM yields a *simple and universally applicable* method to improve episode sampling. To validate that state-of-the-art methods can also benefit from better sampling, we take the recently proposed FEAT [64] algorithm and augment it with our IS-based implementation of online UNIFORM. Concretely, we use their open-source implementation[5] to train both baseline and online UNIFORM sampling. We use the prescribed hyper-parameters without any modifications. Results for ResNet-12 on Mini-ImageNet and Tiered-ImageNet are reported in Table 4, where online UNIFORM outperforms baseline sampling on $3/4$ scenarios and is matched on the remaining one. Thus, better episodic sampling can improve few-shot classification even for the very best methods.

## 6 Conclusion

This manuscript presents a careful study of sampling in the context of few-shot learning, with an eye on episodes and their difficulty. Following an empirical study of episode difficulty, we propose an importance sampling-based method to compare different episode sampling schemes. Our experiments suggest that sampling uniformly over episode difficulty performs best across datasets, training algorithms, network architectures and few-shot protocols. Avenues for future work include devising better sampling strategies, analysis beyond few-shot classification (*e.g.*, regression, reinforcement learning), and a theoretical grounding explaining our observations.

---

[5]Available at: `https://github.com/Sha-Lab/FEAT`

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
