# A  Experimental setup

## A.1  Datasets

We use two standardized few-shot image classification datasets.

*Mini-ImageNet*: This dataset [58] is a subset of ImageNet [10] and consists of $64$ classes for training, $16$ for validation, and $20$ for testing. There are $600$ images per class, with images of size $84 \times 84$. Multiple versions of this dataset exist in the literature; we use the version by Ravi and Larochelle [43].

*Tiered-ImageNet*: A larger subset of ImageNet, Tiered-ImageNet [45] consists of $608$ classes split into $351$, $97$, and $160$ for training, validation, and testing, respectively. Each class has about $1,300$ images of size $84 \times 84$. This dataset ensures that the train, validation, and test classes do not have any semantic overlap and is proposed as a harder few-shot learning benchmark.

We also use the test splits of the following four datasets, as defined by Triantafillou et al. [57].

*CUB-200*: CUB-200 was collected by Welinder et al. [60] and contains $6,033$ bird images classified into $200$ bird species. The original version of the dataset contains $43$ images that are also present in ImageNet. We remove these duplicates to avoid overestimating the transfer capability during evaluation. The test split contains $30$ classes.

*Describable Textures*: Proposed by Cimpoi et al. [8], the task of this dataset is to classify images into $47$ texture classes. Each of the $5,640$ images ($120$ samples per class) contains at least $90\%$ of the class' texture, with sizes between $300 \times 300$ and $640 \times 640$ pixels. The train split has $33$ classes, while validation and test splits both consist of $7$ classes.

*VGG Flowers*: Originally introduced by Nilsback and Zisserman [38], VGG Flowers consists of $102$ flower categories with each category containing between $40$ and $258$ images. While we use Triantafillou et al. [57]'s train ($71$ classes), validation ($15$ classes), and test ($16$ classes) splits, our models operate on the raw images, not the cropped versions.

*FGVC Aircrafts*: Maji et al. [35] introduced this dataset containing $10,200$ images of aircraft partitioned into $102$ classes, each with $100$ samples. The test split contains $15$ classes. As for VGG Flowers, we do not crop those images using bounding box information, thus increasing the classification difficulty.

## A.2  Network architectures

We train two of the most popular network architectures in few-shot learning literature.

*conv(64)$_4$*: This architecture [58] consists of $4$ convolutional layers with $64$ channels per layer.

*ResNet-12*: From the family of deep residual networks [23], this architecture has $4$ blocks, each block constituting $3$ convolutional layers with $64 \times 2^{l-1}$ channels per layer in the $l$'th block. Two versions of this network architecture exist in the literature; we use the one by Oreshkin et al. [39]. The other version by Lee et al. [32] is $1.25\times$ wider and has more parameters.

Both architectures use batch normalization [25] after every convolutional layer with ReLU as the non-linearity. We do not use dropout [54] or any of its variants, like Ghiasi et al. [19]. For MAML and ANIL, a fully-connected layer is appended at the top of the networks.

## A.3  Training algorithms

For the metric-based family, we use ProtoNet with Euclidean [53] and scaled negative cosine similarity measures [20]. Based on the implementation of Gidaris and Komodakis [20], we add a learnable parameter that scales the cosine similarity. Additionally, we use MAML [15] and ANIL [41] as representative gradient-based algorithms. We use the open-source library `lear2learn` [1][6] to implement these algorithms.

---

[6]Available at: `https://github.com/learnables/learn2learn`

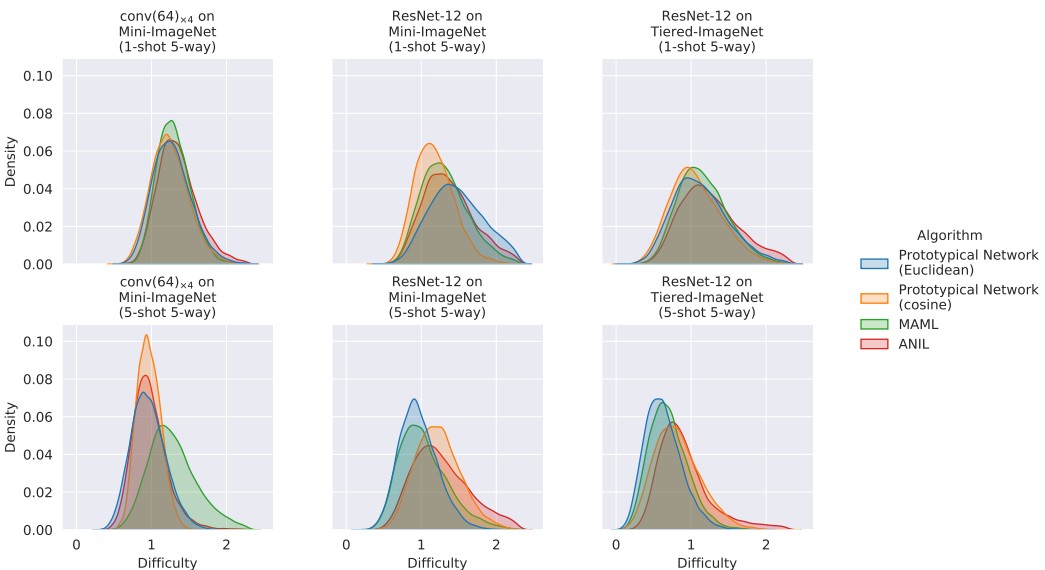

Figure 5: **Episode difficulty is approximately normally distributed - density plots.** Density plots of the episode difficulty computed by conv$(64)_4$'s on Mini-ImageNet (left), ResNet-12's on Mini-ImageNet (center) and ResNet-12's on Tiered-ImageNet (right), trained using ProtoNets (Euclidean and cosine), MAML and ANIL (depicted in the legend). The values are computed over 10k test episodes. The top row is for 1-shot 5-way episodes and the bottom row is for 5-shot 5-way episodes. All the plots follow a bell curve, with the density peak in the middle, which quickly drops-off on either side of the peak.

## A.4   Sampling methods

We compare four sampling methods – EASY, HARD, CURRICULUM, and UNIFORM. In each case, we mimic the target distribution using importance sampling (Section 3.3).

We also have baseline sampling in our comparisons. This involves episodic training without the use of any weighting techniques, hence sampling episodes from the distribution $q(\tau)$ without making any changes to it (Section 2.1). This is the default sampling strategy for few-shot episodic training.

## A.5   Hyper-parameters

We tune hyper-parameters for each algorithm and dataset to work well across different few-shot settings and network architectures. Additionally, we keep the hyper-parameters the same across all different sampling methods for a fair comparison.

All models are trained using ADAM [28] with a learning rate of $10^{-3}$ on a single NVIDIA Tesla V100 GPU. MAML and ANIL use an adaptation learning rate of $0.01$ and $0.1$ respectively, with $5$ adaptation steps taken in both cases. All models are trained for a total of 20k iterations, with a mini-batch of size 16 and 32 for Mini-ImageNet and Tiered-ImageNet respectively. After every 1k iterations, we evaluate on 1k validation episodes. The model with the best validation performance is finally evaluated on 1k test episodes.

## B   Episode difficulty is approximately normally distributed

Sampling episodes from $q(\tau)$ (Section 2.1) induces a distribution over their difficulty $\Omega_{l_\theta}$. Our proposed method estimates this as a normal distribution (Section 3.2), and here we justify why.

We train conv$(64)_4$'s on Mini-ImageNet and ResNet-12's on both Mini-ImageNet and Tiered-ImageNet using baseline sampling. This is done using all four learning algorithms – ProtoNet (Euclidean and cosine), MAML and ANIL – for 1-shot 5-way and 5-shot 5-way classification. We compute the episode difficulty over 10k test episodes, sampled using the episode distribution $q(\tau)$.

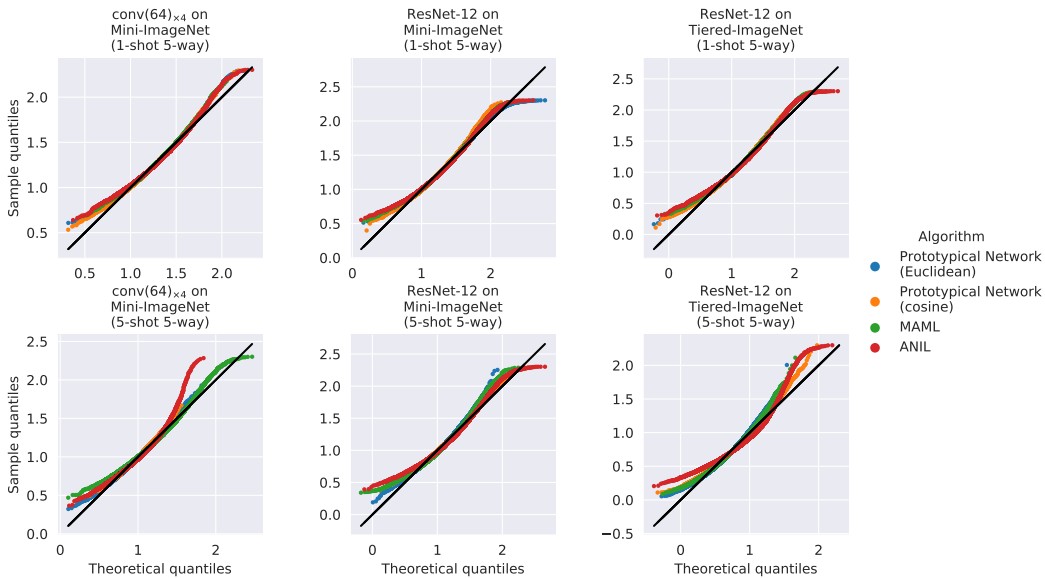

Figure 6: **Episode difficulty is approximately normally distributed - Q-Q plots.** Q-Q plots of the episode difficulty computed by conv$(64)_4$'s on Mini-ImageNet (left), ResNet-12's on Mini-ImageNet (center) and ResNet-12's on Tiered-ImageNet (right), trained using ProtoNets (Euclidean and cosine), MAML and ANIL (depicted in the legend). The values are computed over 10k test episodes and are plotted against normal distributions with the same mean and standard deviation as the episode difficulties. The top row is for 1-shot 5-way episodes and the bottom row is for 5-shot 5-way episodes. We also include the identity line in each plot (in black). The closer the curve is to the identity line, the closer the distribution is to a normal.

Fig. 5 illustrates the density plots of the computed difficulties. We observe that the episode difficulties follow a bell curve in each case, which is naturally modeled with a normal distribution. Fig. 6 includes Q-Q plots for the same, plotted against normal distributions with the same mean and standard deviation as the corresponding episode difficulties. These plots are typically used to assess normality – the closer the curve is to the identity line, the closer the distribution is to a normal, which is observed here.

Table 5: **Episode difficulty is approximately normally distributed - Shapiro-Wilk normality tests.** We compute the episode difficulty for different datasets, algorithms and network architectures, for both the 1-shot 5-way and 5-shot 5-way settings. This is done for 10k test episodes each. In each case, we subsample 50 values 100 times and run the Shapiro-Wilk test on these subsets (with $\alpha = 0.05$). The rejection rates of the null hypothesis are averaged over everything but the axes mentioned in the left column and are mentioned in the column on the right. The average rejection rate does not exceed 20%.

|  |  | Rejection rate (%) |
|---|---|---|
| Dataset | Mini-ImageNet | 14.25 |
|  | Tiered-ImageNet | 17.38 |
| Shots | 1-shot | 14.17 |
|  | 5-shot | 16.42 |
| Algorithm | ProtoNet (Euclidean) | 19.67 |
|  | ProtoNet (cosine) | 09.17 |
|  | MAML | 13.33 |
|  | ANIL | 19.00 |
| Network Architecture | conv$(64)_4$ | 09.63 |
|  | ResNet-12 | 18.13 |

We additionally run the Shapiro-Wilk test for normality [50] on the computed episode difficulties, which tests for the null hypothesis that the data is drawn from a normal distribution. The p-value for this test is sensitive to the sample size – for large sample sizes, trivial departures from the normal

distribution can be detected, making the p-values unreliable. Instead, we subsample 50 values 100 times and run the test on these subsets (with $\alpha = 0.05$). Table 5 summarizes the rejection rates of the null hypothesis averaged over datasets, shots, algorithms and network architectures. Regardless of which axis the rejection rate is averaged over, it does not exceed $20\%$. These results suggest that our assumption of estimating the induced distribution over the episode difficulty as a normal distribution is plausible.

## C    Episode difficulty is independent from modeling choices

This section provides the full version of the figures from Section 5.2.2.

Fig. 7 reports correlation plots for the difficulty of episodes when measured with two different architectures. We use $conv(64)_4$ and ResNet-12's trained on Mini-ImageNet (1-shot 5-way) with all training algorithms to compute episode difficulties for 10k test episodes. We then compute the Spearman rank-order correlation coefficients across the two architectures, for a given algorithm. The correlation coefficients are $0.57$ for ProtoNet (Euclidean), $0.72$ for ProtoNet (cosine), $0.58$ for MAML, and $0.49$ for ANIL. As mentioned in Section 5.2.2, this positive correlation suggests that episode difficulty is transferred across network architectures with high probability.

Fig. 8 tracks the difficulty of easy and hard episodes over training iterations, for all four training algorithms. Out of 1k test episodes, we select the 50 *easiest* and 50 *hardest* episodes, *i.e.*, episodes with the lowest and highest difficulties respectively. We measure difficulty on these episodes every 1k training iterations, and observe that the difficulty lines for easy and hard episodes never cross – easy episodes remain easy and hard episodes remain hard. This observation suggests that episode difficulty transfers across different model parameters during training, justifying our online estimation of difficulty parameters $\mu$ and $\sigma^2$ (Section 3.2).

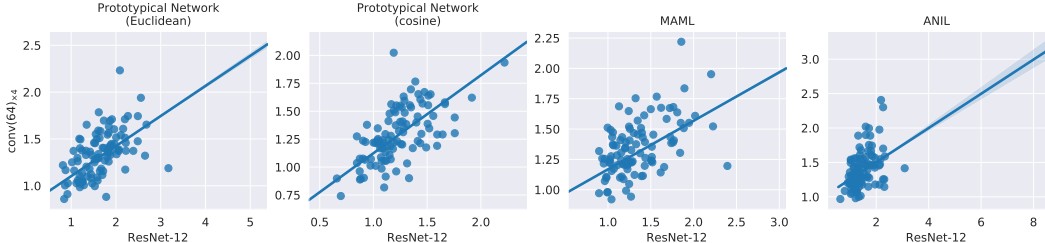

Figure 7: **Episode difficulty transfers across network architectures.** Scatter plots (with regression line and $95\%$ confidence interval) of episode difficulties computed on 10k 1-shot 5-way Mini-ImageNet episodes by $conv(64)_4$'s and ResNet-12's trained using different training algorithms. Similar to Fig. 3, we observe that an episode that is hard for one architecture is very likely to be hard for another, for all training algorithms.

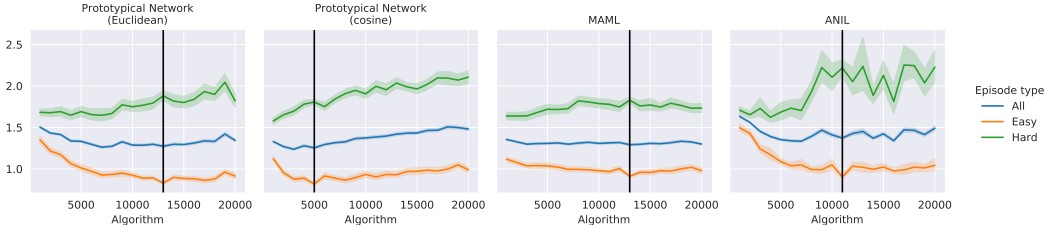

Figure 8: **Episode difficulty transfers across model parameters during training.** Similar to Fig. 4, out of 1k test episodes, we select the 50 easiest and 50 hardest episodes and track the difficulties of them all throughout training. The average difficulty of the episodes decreases over time, until convergence (vertical line), after which the model overfits. Additionally, easier episodes remain easy while harder episodes remain hard, indicating that episode difficulty transfers from one set of parameters to the next. Note that since we validate the models every 1k iterations, these plots are not continuous and do not contain the values for the first 1k training iterations (during which the episode difficulty drops quickly).

Table 6: **Few-shot accuracies on benchmark datasets for 5-way few-shot episodes in the offline setting.** The mean accuracy and the 95% confidence interval are reported for evaluation done over 1k test episodes. The first row in every scenario denotes baseline sampling. Best results for a fixed scenario are shown in bold. Results where a sampling technique is better than or comparable to baseline sampling are denoted by †. Overall, UNIFORM is among the best sampling methods in 19/24 scenarios.

| | Mini-ImageNet | | | | Tiered-ImageNet | |
| --- | --- | --- | --- | --- | --- | --- |
| | conv(64)$_4$ | | ResNet-12 | | ResNet-12 | |
| | 1-shot (%) | 5-shot (%) | 1-shot (%) | 5-shot (%) | 1-shot (%) | 5-shot (%) |
| ProtoNet (Euclidean) | **49.06±0.60** | 65.28±0.52 | 49.67±0.64 | 67.45±0.51 | **59.10±0.73** | 76.95±0.56 |
| + EASY | **48.83±0.61**† | 65.92±0.55† | 51.08±0.63† | 67.30±0.52† | 57.68±0.75 | **78.10±0.53**† |
| + HARD | 45.69±0.61 | **66.47±0.52**† | 52.50±0.62† | **71.03±0.51**† | 54.85±0.71 | 76.15±0.56 |
| + CURRICULUM | 48.23±0.63 | 65.77±0.51† | 50.00±0.61† | 70.49±0.51† | **59.15±0.76**† | 78.25±0.53† |
| + UNIFORM | 48.19±0.62 | 66.73±0.52† | **53.94±0.63**† | 70.79±0.49† | 58.63±0.76† | 78.62±0.55† |
| ProtoNet (cosine) | **50.03±0.61** | 61.56±0.53 | 52.85±0.64 | 62.11±0.52 | **60.01±0.73** | 72.75±0.59 |
| + EASY | 49.60±0.61† | 65.17±0.53† | 53.35±0.63† | 63.55±0.53† | 60.03±0.75† | 74.65±0.57† |
| + HARD | 49.01±0.60 | **66.45±0.50**† | 52.65±0.63† | 70.15±0.51† | 55.44±0.72 | 75.97±0.55† |
| + CURRICULUM | 49.38±0.61 | 64.12±0.53† | 53.21±0.65† | 65.89±0.52† | **60.37±0.76**† | 75.32±0.58† |
| + UNIFORM | **50.07±0.59**† | 66.33±0.52† | **54.27±0.65**† | 70.85±0.51† | 60.27±0.75† | **78.36±0.54**† |
| MAML | **46.88±0.60** | 55.16±0.55 | 49.92±0.65 | 63.93±0.59 | **55.37±0.74** | **72.93±0.60** |
| + EASY | 44.52±0.60 | 57.36±0.59† | 51.62±0.67† | 64.33±0.61† | 53.39±0.79 | 69.81±0.68 |
| + HARD | 42.93±0.61 | 60.42±0.55† | 49.57±0.69† | **66.93±0.55**† | 50.48±0.73 | 71.20±0.63 |
| + CURRICULUM | 45.42±0.60 | **61.61±0.55**† | **52.21±0.67**† | 66.25±0.60† | 54.13±0.77 | 71.47±0.63 |
| + UNIFORM | **46.67±0.63**† | 62.09±0.55† | 52.65±0.65† | 66.76±0.57† | 54.58±0.77 | 72.00±0.66 |
| ANIL | **46.59±0.60** | **63.47±0.55** | **49.65±0.65** | 59.51±0.56 | 54.77±0.76 | 69.28±0.67 |
| + EASY | 44.83±0.63 | 62.23±0.56 | 49.40±0.64† | 56.73±0.60 | 54.50±0.80† | 65.45±0.66 |
| + HARD | 43.30±0.58 | 59.87±0.55 | 47.91±0.62 | 62.05±0.59† | 50.22±0.71 | 62.06±0.65 |
| + CURRICULUM | 45.69±0.60 | **63.00±0.54**† | 50.22±0.66† | 61.76±0.57† | **55.59±0.78**† | **69.83±0.73**† |
| + UNIFORM | **46.93±0.62**† | 62.75±0.60 | 49.56±0.62† | **64.72±0.60**† | 54.15±0.79† | 70.44±0.69† |

# D   Comparing episode sampling methods

In addition to the discussion in Section 5.3, this section presents the full suite of results for the comparison of different episode sampling methods. We compute results over 2 datasets, 2 network architectures, 4 algorithms and 2 few-shot protocols, resulting in 24 total scenarios. Table 6 contains all performance numbers. As mentioned in the main text, UNIFORM is among the better sampling schemes in 19/24 scenarios, followed by baseline sampling which is competitive in 10/24 scenarios. Importantly, when UNIFORM underperforms it is a close second.

# E   Difference in effectiveness in the 1- and 5-shot settings

The 1-shot setting is inherently noisier than 5-shot. Support samples are randomly drawn from the class-populations, which are then used to construct the few-shot classifier. Sampling only 1 support per-class is more susceptible to outliers in the query set than sampling 5 (the higher the support-shot, the better the estimate of the class-population). This noise propagates to the loss (in the case of baseline sampling) as well as the weighted loss (in the case of UNIFORM sampling). Hence, larger noise degrades the approximation to a uniform distribution over episode difficulty and ultimately results in UNIFORM not getting as much gain in the 1-shot setting.

We empirically confirm this hypothesis. We use the same 24 scenarios as the ones in Sections 5.3 and 5.4 and compare the training procedures of UNIFORM under 1- vs. 5-shot settings. Using Eq. (3), we compute the per-episode weighted loss during the training process, followed by the per-mini-batch standard deviation. The average deviation is higher under the 1-shot than the 5-shot setting in all scenarios (for both offline and online settings). Additionally, the average deviation is ≈ 1.9 times larger under the 1-shot setting. These experiments confirm the above hypothesis and help explain

why UNIFORM (online) outperforms the baseline in (only) 5/12 1-shot scenarios, is comparable in 6/12, and underperforms in 1/12.

Table 7: **Few-shot accuracies on benchmark datasets after training on Mini-ImageNet for 5-way few-shot episodes in the offline and online settings.** The mean accuracy and the 95% confidence interval are reported for evaluation done over 1,000 test episodes. Best results for a fixed scenario are shown in bold. The first row in every scenario denotes baseline sampling. Compared to baseline sampling, online UNIFORM does statistically better in 49/64 scenarios, comparable in 12/64 scenarios and worse in only 3/64 scenarios.

| | conv(64)$_4$ | | ResNet-12 | |
|---|---|---|---|---|
| | 1-shot (%) | 5-shot (%) | 1-shot (%) | 5-shot (%) |
| CUB-200 | | | | |
| ProtoNet (Euclidean) | **37.24**±**0.53** | 52.07±0.53 | 36.53±0.54 | 51.49±0.56 |
| + UNIFORM (Online) | **37.08**±**0.53** | **53.32**±**0.53** | **39.48**±**0.56** | **56.57**±**0.55** |
| ProtoNet (cosine) | 37.49±0.54 | 49.31±0.53 | 38.67±0.60 | 49.75±0.57 |
| + UNIFORM (Online) | **41.56**±**0.58** | **54.17**±**0.53** | **40.55**±**0.60** | **56.30**±**0.55** |
| MAML | 34.52±0.53 | **47.11**±**0.60** | 35.80±0.56 | 45.16±0.62 |
| + UNIFORM (Online) | **35.84**±**0.54** | **46.67**±**0.55** | **37.18**±**0.55** | **46.58**±**0.58** |
| ANIL | 35.40±0.54 | 38.20±0.56 | 33.20±0.54 | 39.26±0.58 |
| + UNIFORM (Online) | **36.89**±**0.55** | **42.83**±**0.58** | **34.47**±**0.56** | **42.08**±**0.58** |
| Describable Textures | | | | |
| ProtoNet (Euclidean) | 32.05±0.45 | **45.03**±**0.44** | 31.87±0.45 | 44.10±0.43 |
| + UNIFORM (Online) | **32.69**±**0.49** | **45.23**±**0.43** | **33.55**±**0.46** | **47.37**±**0.43** |
| ProtoNet (cosine) | 32.09±0.45 | 38.44±0.41 | 31.48±0.45 | 39.46±0.41 |
| + UNIFORM (Online) | **33.63**±**0.47** | **43.28**±**0.44** | **32.69**±**0.48** | **45.56**±**0.42** |
| MAML | 29.47±0.46 | 37.85±0.47 | **32.19**±**0.48** | 41.14±0.46 |
| + UNIFORM (Online) | **31.84**±**0.49** | **40.81**±**0.44** | 31.65±0.46 | **43.21**±**0.44** |
| ANIL | 29.86±0.46 | 40.69±0.46 | 28.85±0.41 | 37.04±0.44 |
| + UNIFORM (Online) | **31.29**±**0.48** | **41.42**±**0.45** | **31.38**±**0.47** | **39.03**±**0.47** |
| FGVC-Aircraft | | | | |
| ProtoNet (Euclidean) | **26.03**±**0.37** | 39.41±0.48 | 25.98±0.39 | 36.76±0.45 |
| + UNIFORM (Online) | **26.18**±**0.38** | **40.23**±**0.46** | **27.43**±**0.42** | **38.49**±**0.46** |
| ProtoNet (cosine) | **27.11**±**0.39** | 32.14±0.38 | 25.23±0.39 | 32.07±0.41 |
| + UNIFORM (Online) | **27.15**±**0.38** | **37.78**±**0.45** | **26.89**±**0.39** | **37.42**±**0.44** |
| MAML | **26.78**±**0.38** | **34.21**±**0.41** | 25.50±0.39 | 29.38±0.40 |
| + UNIFORM (Online) | **26.62**±**0.39** | **34.41**±**0.44** | **26.22**±**0.39** | **30.21**±**0.43** |
| ANIL | **25.67**±**0.37** | 27.17±0.36 | 23.27±0.31 | 24.52±0.29 |
| + UNIFORM (Online) | **25.60**±**0.37** | **27.92**±**0.39** | **23.78**±**0.34** | **28.70**±**0.39** |
| VGG Flowers | | | | |
| ProtoNet (Euclidean) | 53.50±0.63 | 70.96±0.51 | **57.74**±**0.68** | 74.87±0.49 |
| + UNIFORM (Online) | **54.72**±**0.65** | **73.59**±**0.49** | 55.94±0.67 | **76.62**±**0.50** |
| ProtoNet (cosine) | 52.94±0.62 | 66.04±0.53 | 52.98±0.65 | 66.79±0.51 |
| + UNIFORM (Online) | **54.23**±**0.63** | **71.93**±**0.48** | **57.06**±**0.65** | **67.31**±**0.48** |
| MAML | **49.70**±**0.60** | **63.69**±**0.54** | **50.13**±**0.64** | 61.41±0.63 |
| + UNIFORM (Online) | **49.72**±**0.60** | **63.52**±**0.54** | 49.53±0.65 | **63.99**±**0.58** |
| ANIL | **47.03**±**0.65** | 46.40±0.66 | **42.05**±**0.67** | 40.01±0.65 |
| + UNIFORM (Online) | **47.48**±**0.67** | **47.08**±**0.67** | 38.94±0.61 | **50.25**±**0.63** |

## F   Better sampling improves cross-domain few-shot classification

In Section 5.5, we show that few-shot performance in the cross-domain setting can benefit from better sampling. We train models on Mini-ImageNet (as done in Section 5.4) and test the few-

shot performance on the following datasets: CUB-200 [60], Describable Textures [8], FGVC-Aircraft [35], VGG Flowers [38]. We use conv$(64)_4$ and ResNet-12 network architectures trained using ProtoNet (Euclidean and cosine), MAML and ANIL algorithms for the 5-ways 1- and 5-shot settings. Altogether, these makeup $64$ new scenarios. We measure the accuracy on the test splits of [57].

We compare online UNIFORM against baseline sampling and observe that online UNIFORM does statistically better in $49/64$ scenarios, comparable in $12/64$ scenarios, and worse in only $3/64$ scenarios. The performance numbers are included in Table 7.

This further goes to show that sampling under the episodic training paradigm matters. Using online UNIFORM leads to statistically significant improvements over the ubiquitous baseline sampling in most cases and rarely degrades performance.

## G  Number of trials

In Tables 2 and 6 we make use of one random seed to give one training job per scenario per sampling method. However we report performances over 1k test episodes, as is typically done in few-shot learning. We additionally ran 3 training jobs for baseline sampling and online UNIFORM, resulting in 3 training jobs per scenario per sampling method. We observe that the difference in accuracy is .20% and .02% on average (ignoring the standard deviations) for baseline sampling and online UNIFORM; the effect of multiple random seeds is diminished when testing over many episodes.