# OpenReview forum: "Uniform Sampling over Episode Difficulty"
_NeurIPS.cc/2021/Conference — NeurIPS 2021 Spotlight_

### Official Review · Reviewer_sGro · 2021-07-16

**Rating:** 8
**Confidence:** 4

**Summary:**

This paper explores episode/task difficulty in few-shot learning. The task difficulty is defined as the negative log-likelihood over the query set given the support set and model parameters. The authors find evidence that episode difficulty likely follows a normal distribution and proposes a new method, called UNIFORM, based on importance sampling. The new method achieves similar or better performance than other sampling procedures (EASY, HARD, and CURRICULUM). The paper also presents two variants of UNIFORM: offline and online.

**Limitations And Societal Impact:**

Limitations not given.

**Main Review:**

\+ means strength, \- means weakness

**Originality**

\+ *Interesting and novel insights into the task difficulty for meta-learners.* Although the idea of controlling episode difficulty is not very novel [47, 28], the authors go about it in a novel way and propose to use importance sampling for a faster and more efficient way of sampling episodes that is model agnostic.

**Quality**

\+ *Solid evaluation and analysis.* Testing over multiple models, algorithms, and benchmarks makes for a convincing set of results.

\+ *Confirmed assumptions through experimentation.* The paper uses Q-Q plots and Shapiro-Wilk to verify the assumption that episode difficulty follows a normal distribution.

\- *Advantage of UNIFORM over other procedures is not consistent.* The tables show that UNIFORM does not always offer a clear advantage over the results, especially in the 1-shot setting. Do the authors have a theory for why the method is not as effective on the 1 shot setting?

**Clarity**

\+ *Experiments are well designed, and the results are clear.*

\+ *Paper is organized and well-written.*

**Significance**

\+ *Interesting solution to a relevant problem.* Episode difficulty is understudied in few-shot learning, and uniform presents a model agnostic solution to the problem of how to sample tasks during meta-learning.


_________
## Post-Rebuttal
_________

I have read the rebuttal and other reviews, and I increase my original rating. This paper addresses an interesting and relevant issue of episode difficulty in meta-learning and will be a particularly valuable contribution at the conference. The paper presents a solid set of evaluations and analyses, verifying assumptions (e.g. via Q-Q plots and Shapiro-Wilk tests) and reporting results across multiple benchmarks. The paper presents a novel, simple but effective method that, on the most part, boosts algorithm performance by a couple of additional accuracy points on average. Therefore, I recommend this work for the conference with a score of 8.

**Time Spent Reviewing:**

1.5

---

> ### Author Response · Authors · 2021-08-10
> **Response to Reviewer sGro**
>
> We thank Reviewer **sGro** for the time taken in reviewing our submission and the encouraging feedback. We reply to the concern below.
>
> **Why is UNIFORM not as effective in the 1-shot setting?**
>
> The 1-shot setting is inherently noisier than 5-shot. Support samples are randomly drawn from the class-populations, which are then used to construct the few-shot classifier. Sampling only 1 support per-class is more susceptible to outliers in the query set than sampling 5 (the higher the support-shot, the better the estimate of the class-population). This noise propagates to the loss (in the case of baseline sampling) as well as the weighted loss (in the case of UNIFORM). Hence, larger noise degrades the approximation to a uniform distribution over episode difficulty and ultimately results in UNIFORM not getting as much gain in the 1-shot setting.
>
> We empirically confirm this hypothesis. We use the same 24 scenarios as the ones in the paper and compare the training procedures of UNIFORM under 1- vs. 5-shot settings. Using Eq. 3, we compute the per-episode weighted loss during the training process, followed by the per-mini-batch standard deviation. The average deviation is higher under the 1-shot than the 5-shot setting in all scenarios (for both offline and online settings). Additionally, the average deviation is ~1.9 times larger under the 1-shot setting. These experiments confirm the above hypothesis and help explain why UNIFORM (online) outperforms the baseline in (only) 4/12 scenarios, is comparable in 7/12, and underperforms in 1/12.
>
> We thank the reviewer for their clarification request, and will add this new discussion to the limitation of our IS-based uniform sampler.

---

> > ### Comment · Reviewer_sGro · 2021-08-24
> > **Thanks for the response.**
> >
> > I thank the authors for the clarifications and the additional experiments. I am inclined to keep my original score with increased confidence.

---

### Official Review · Reviewer_KvDG · 2021-07-16

**Rating:** 6
**Confidence:** 3

**Summary:**

Meta-learning (few-shot learning) algorithms, in general, equally treat all tasks for the training process. Tasks are sampled uniformly at random. This paper proposes a novel meta-learning framework that controls the task sampling probabilities according to difficulties. This paper measures the difficulties of tasks using the negative log-likelihood and leverages the difficulties to introduce different weights to tasks. The main idea of the learning framework is that the prior of the difficulties follows a normal distribution while the target sampling distribution could be uniform or some other interesting forms. Then, important sampling provides different weights to different tasks. From experiments, the uniform target distribution shows good results.

**Limitations And Societal Impact:**

This paper did not properly discuss their limitation. Please refer to the main review comments.

**Main Review:**

Originality:
The paper proposes a novel task sampling framework for meta-learning, which is, indeed, a task weighting method considering task difficulties. The authors should discuss other existing task weighting methods.

Quality:
The proposed episodic training method is intuitive, and the performance looks good.

The proposed algorithm is evaluated with ProtoNet, MAML, and ANIL.

Although the training algorithms are well-known algorithms, they are not the state-of-the-art. Since one of the claims is that the proposed method is algorithm agnostic, it is necessary to perform experiments with recent training algorithms.

The authors only consider Mini-ImageNet and Tiered-ImageNet for the evaluation. In meta-learning research, specific data sets such as CUB and Cars are also used frequently, and cross-domain tasks are also important. More experiments with various data sets would be very useful.

Although the performance shows great results, it is not clear why uniform sampling outperforms other sampling distributions. The choice of the negative log-likelihood as the difficulty measure also is required to be justified. What will happen when we use other difficulty measures?

Clarity:
This paper is easy to read.

Significance:
This paper proposes a novel episodic learning framework. The idea could be used to build other research results.


**Time Spent Reviewing:**

15

---

> ### Author Response · Authors · 2021-08-10
> **Response to Reviewer KvDG**
>
> We thank Reviewer **KvDG** for the  time spent in reviewing our submission and the feedback. We address the concerns below.
>
> **Discussion of other episode-weighting methods.**
>
> Indeed, many prior works can be understood as weighting episodes; however, few discuss the implications of this weighting on the underlying episode distribution, which is the focus of our submission.
>
> For example, [1] and [2] propose to meta-learn episode-weighting parameters but, crucially, their setting is different from ours as the meta-learner can rely on an additional set of validation episodes. Importantly, they do not discuss the effect of the meta-learned episode weights on the sampling distribution. From another line of work, [3] argues that MAML implicitly upweights more difficult episodes, akin to HARD sampling, but their analysis focuses on understanding the differences between MAML and ERM. Finally, other approaches such as active meta-learning [4, 5] are outside the scope of our work, since we assume all episodes are available at the beginning of training (which is standard in few-shot classification).
>
> **Results for SOTA few-shot algorithms.**
>
> We include results for FEAT (c.f., Table 4, p. 9) which is comparable to or better than other state-of-the-art methods. Our conclusions carry over to FEAT since UNIFORM improves few-shot accuracy in 3/4 scenarios and matches performance in the other. Note that for these experiments, we used the author's open-source implementation and their prescribed hyper-parameters, without any additional tuning.
>
> **Results for cross-domain few-shot learning.**
>
> As suggested by the reviewer, we train models on Mini-ImageNet and transfer them to the following datasets: CUB-200 [8], Describable Textures [9], FGVC-Aircraft [10], VGG Flowers [11]. We measure accuracy on the test splits of [12]. We use conv$(64)_4$ and ResNet-$12$ architectures trained using ProtoNet (Euclidean and cosine), MAML and ANIL algorithms for the 5-ways 1- and 5-shot settings. Altogether these make up 64 new scenarios.
>
> We compare UNIFORM (online) against baseline sampling and observe that UNIFORM does statistically better in 49/64 (76.56%) scenarios, comparable in 12/64 (18.75%) scenarios and worse in only 3/64 (4.69%) scenarios. We include a snippet of these results in the table below.
>
> These results further go to show that sampling under the episodic training paradigm matters and that UNIFORM can boost performance over baseline sampling. Using UNIFORM leads to statistically significant improvements over the ubiquitous baseline sampling in most cases and rarely degrades performance.
> We thank the reviewer for these suggested cross-domain experiments and will add them to the main text.
>
> |                      | CUB-Res12-1shot | CUB-Res12-5shot | Textures-CNN64-1shot | Textures-CNN64-5shot |
> |:---------------------|:---------------:|:---------------:|:--------------------:|:--------------------:|
> | ProtoNet (Euclidean) |   36.53±0.54    |   51.49±0.56    |      32.05±0.45      |    **45.03±0.44**    |
> | \+ UNIFORM (Online)  | **39.48±0.56**  | **56.57±0.55**  |    **32.69±0.49**    |    **45.23±0.43**    |
> | ProtoNet (cosine)    |   38.67±0.60    |   49.75±0.57    |      32.09±0.45      |      38.44±0.41      |
> | \+ UNIFORM (Online)  | **40.55±0.60**  | **56.30±0.55**  |    **33.63±0.47**    |    **43.28±0.44**    |
> | MAML                 |   35.80±0.56    |   45.16±0.62    |      29.47±0.46      |      37.85±0.47      |
> | \+ UNIFORM (Online)  | **37.18±0.55**  | **46.58±0.58**  |    **31.84±0.49**    |    **40.81±0.44**    |
> | ANIL                 |   33.20±0.54    |   39.26±0.58    |      29.86±0.46      |      40.69±0.46      |
> | \+ UNIFORM (Online)  | **34.47±0.56**  | **42.08±0.58**  |    **31.29±0.48**    |    **41.42±0.45**    |
>
> **Table caption:** Few-shot accuracies on CUB-200 and Describable Textures after training on Mini_imageNet for 5-way few-shot episodes. This is a snippet of a larger scale experiment. The mean accuracy and the 95% confidence interval are reported for evaluation done over 1,000 test episodes. Best results for a fixed scenario are shown in bold. The first row in every scenario denotes baseline sampling. UNIFORM does statistically better in 15/16 scenarios and comparable in 1/16 scenarios.
>
>
> **Why does uniform sampling outperform other sampling schemes?**
>
> Our intuition is that uniform sampling puts a uniform distribution prior over the (unseen) test episodes, with the intention of performing well across the entire difficulty spectrum. This acts as a regularizer, forcing the model to be equally discriminative for easy and hard episodes. This uninformative prior is the safest choice without additional information about test episodes, for other sampling distributions implicitly favor some episodes over others. In fact, if we knew the test episode distribution, upweighting train episodes that are most likely under that distribution will improve transfer accuracy. This argument can be formalized as shown in Remark 2 of [6]. We will expand on this point in the main text.
>
> **Why use negative log-likelihood (NLL) as a measure of episode difficulty?**
>
> We used the NLL as a measure of difficulty because it is readily available during training, and does not require extra computation (unlike other proxy measures, such as gradient norm). Moreover, it is closely connected to the average log-odds, which was used as a difficulty measure for the analysis in [7]. Finally, we also considered using accuracy as a measure of difficulty. However, accuracy is ill-suited for the few-shot setting (difficulty values become discretized) and requires different modeling assumptions (due to bounded support on [0, 1]). We include more details in our response to Reviewer **3S5L**.
>
> One caveat is that the above difficulty measures, including NLL, are entangled with modeling choices such as training algorithm and architecture. For this reason, Section 5.2 validates our modeling assumptions and shows that relative difficulty computed using NLL is (mostly) invariant to such modeling choices.
>
>
> **References**
>
> 1. Killamsetty et al., "A Reweighted Meta Learning Framework for Robust Few Shot Learning", arXiv 2020.
> 2. Aimen et al., "A Task Attended Meta-Learning for Few-Shot Learning", arXiv 2021.
> 3. Collins et al., "How Does the Task Landscape Affect MAML Performance?", arXiv 2020.
> 4. Kaddour et al., "Probabilistic Active Meta-Learning", NeurIPS 2020.
> 5. Ermis et al., "Towards Robust Episodic Meta-Learning", UAI 2021.
> 6. Fallah et al., "Generalization of Model-Agnostic Meta-Learning Algorithms: Recurring and Unseen Tasks", arXiv 2021.
> 7. Dhillon et al., "A Baseline for Few-Shot Classification", ICLR 2020.
> 8. Welinder et al., "Caltech-UCSD Birds 200", Caltech Technical Report 2010.
> 9. Cimpoi et al., "Describing Textures in the Wild", CVPR 2014.
> 10. Maji et al., "Fine-Grained Visual Classification of Aircraft", arXiv 2013.
> 11. Nilsback and Zisserman, "A Visual Vocabulary for Flower Classification", CVPR 2006.
> 12. Triantafillou et al., "Meta-Dataset: A Dataset of Datasets for Learning to Learn from Few Examples." ICLR 2020.

---

> > ### Comment · Reviewer_KvDG · 2021-08-19
> > **Re**
> >
> > Thanks for your response and additional experiments. Your answer helps me understand your paper much more clearly.

---

### Official Review · Reviewer_3S5L · 2021-07-16

**Rating:** 8
**Confidence:** 4

**Summary:**

This paper investigates the effect of episode difficulty and sampling schemes on the performance of few-shot classification algorithms. It defines difficulty based on the negative log predictive likelihood of the few-shot learner and proposes to use importance sampling to mimic various sampling strategies. Experiments demonstrate that episodic difficulty is (a) approximately normally distributed, (b) tends to transfer across architectures and learning algorithms, (c) tends to be consistent over the course of training, and (d) that a uniform sampling scheme over difficulty tends to produce the best results. An online scheme for computing importance weights is also proposed.


=== Post-rebuttal ===

I have read the authors' response and other reviews and will maintain my original rating.

**Ethical Concerns:**

No ethical issues.

**Limitations And Societal Impact:**

This work proposes a general purpose improvement to episodically trained few-shot learning algorithm and therefore the same societal implications that result from few-shot learning also apply here.

**Main Review:**

Episodic training underpins many few-shot learning algorithms and thus understanding it well is a key concern. For this reason, the results presented in this paper will be of broad interest to the few-shot learning community. The proposed online uniform difficulty sampling scheme is simple and therefore should be relatively easy to implement.

The paper is very well-written and easy to read. Experiments are systematic and easily interpreted. Though the performance gains from uniform difficulty sampling are not always very large, it does appear to be consistently better than other approaches.

I would have liked to see a discussion of future extensions, for example training a network to directly estimate sample difficulty or possibly a policy network that outputs the importance weight to be assigned to an episode. Also it would be good to understand what the authors tried that did not work. For example, were other measures of episode difficulty such as accuracy considered? Were other sampling schemes tried and discarded?

Overall I believe this is a strong paper and that others will benefit from reading it.

**Time Spent Reviewing:**

1

---

> ### Author Response · Authors · 2021-08-10
> **Response to Reviewer 3S5L**
>
> We thank Reviewer **3S5L** for the review encouraging further understanding of episodic sampling. We respond to the questions below.
>
> **Discussion of future extensions.**
>
> As pointed out by the reviewer, one such avenue is designing better episodic sampling methods to accelerate convergence or improve few-shot performance. Another consists of extending our analysis beyond few-shot classification, for example, to regression or reinforcement learning tasks. Finally, our analysis is mostly empirical and could be formalized. We note that concurrent work [1] presents preliminary work in this direction, characterizing the population risk on test episodes when the distribution over train episodes is uniform (c.f., their Th. 3).
> Following this suggestion, we will add a paragraph outlining avenues for future work to the conclusion.
>
> **What did we try that did not work?**
>
> We considered using accuracy as a measure of episode difficulty since it is also readily available during training. However, accuracy is ill-suited to the few-shot setting as it artificially discretizes the difficulty of an episode. For example, in the 5-ways 1-query-shot case, only 1 of 6 accuracy values are permissible, i.e., {0%, 20%, 40%, 60%, 80%, 100%}. Additionally, accuracy only admits support on [0, 1] which asks for different modeling assumptions that come with their own caveats; for example, the beta distribution, requires $\sigma^2 < \mu (1 - \mu)$ for the density to be estimated from mean $\mu$ and variance $\sigma^2$.
> We chose to focus on the negative log-likelihood to sidestep these difficulties, which led to simple yet efficient modeling assumptions.
>
> **References**
>
> 1. Fallah et al., "Generalization of Model-Agnostic Meta-LearningAlgorithms: Recurring and Unseen Tasks", arXiv 2021.

---

> > ### Comment · Reviewer_3S5L · 2021-09-01
> > **Re: Response**
> >
> > Thanks to the authors for their response. My concerns have been addressed and I am willing to maintain my original recommendation of acceptance.

---

### Official Review · Reviewer_wemo · 2021-07-22

**Rating:** 6
**Confidence:** 4

**Summary:**

This submission studies episodic sampling, which is used to sample datasets (consisting of support and query sets) for training meta-learning models. The authors define a notion of episodic difficulty (the negative log-likelihood of the query set given the support set and model parameters) and propose using importance sampling to train using several different types of episode-sampling strategies for meta-training. By using importance sampling, there is no sampling required for the target distribution but just weighting the episode loss based on the proposal and target distribution. They conduct experiments both to confirm some of their assumptions that are critical for their proposal (that episodic difficulty is normally distributed) and to show how their episodic sampling strategies compare against random episodic sampling, which is the dominant strategy used by work in this area.


**Ethical Concerns:**

No ethical concerns.

**Limitations And Societal Impact:**

Though not discussed, the societal impact of this work is probably similar to that of any other few-shot method.

**Main Review:**

Originality: the paper considers an under-studied but important aspect of meta-learning that to my knowledge hasn't been studied in previous work. Rather than modify an existing architecture or method for meta-learning, the authors are studying an interesting aspect of the problem that most work in this area take for granted.

Quality: authors conducted experiments beyond simply comparing methods on benchmarks to confirm assumptions made in the paper. These include confirming the normal distribution of episodic difficulty or that episodic difficulty is independent from modeling choice. Additionally, benchmark experiments were conducted for a variety of networks, datasets, and meta-learning methods.

Clarity: the paper is well-written - the ideas are presented clearly and the description of the proposed method and experiments are easy to understand.

Significance: my main concern with the paper is the experimental results. I believe the sampling strategies proposed here would be used by the community if the benefits they offer are significant. I believe the online strategy is much more feasible for use in training but I don't believe it's clear it offers a big benefit in performance. For example, in 1/3 of the scenarios presented for mini-ImageNet and tiered-ImageNet, the baseline sampling strategy is better or equivalent in terms of performance. Additionally, for the FEAT comparison, the proposed strategy offers basically the same performance as the baseline one. Thus, because of the work involved in implementing the proposed sampling strategies, I'm not sure the benefit shown is large or consistent enough for this to become something standard that is used in this research subfield.

**Time Spent Reviewing:**

3

---

> ### Author Response · Authors · 2021-08-10
> **Response to Reviewer wemo**
>
> We thank Reviewer **wemo** for the time spent in reviewing our submission and the detailed feedback. We respond to the concerns below.
>
> **Improvements of UNIFORM are not significant enough.**
>
> UNIFORM increases accuracy over baseline sampling on average by +2.39% for the offline formulation (min: -1.44%, max: +8.74%) and +1.85% for the online one (min: -1.26%, max: +6.67%). These improvements are in line with changing the few-shot algorithm from ProtoNet to MetaOptNet (+2.35%; c.f. Table 3 of [1]).
> Improvements for FEAT are less impressive but they are statistically significant. They also come "for free" since we did not re-tune hyper-parameters in the author's open-source implementation. Finally, UNIFORM (online) stands out in the new cross-domain experiments as it is statistically better in 49/64 (76.56%) scenarios, comparable in 12/64 (18.75%), and worse in only 3/64 (4.69%) (we include more details in our response to Reviewer **KvDG**).
>
> As underlined in the first half of the review, we thoroughly analyzed an "under-studied but important aspect of meta-learning", "that most work in this area take for granted".
> Our proposed IS-based scheme is algorithm-agnostic, conceptually simple, and implemented in fewer than 100 lines of code (c.f. Appendix H).
> We believe this low implementation to improvement trade-off makes it a compelling baseline and hope our results can motivate further research in episode sampling.
>
>
> **References**
>
> 1. Lee et al., "Meta-Learning with Differentiable Convex Optimization", CVPR 2021.

---

> > ### Comment · Reviewer_wemo · 2021-09-01
> > **Thanks for the response**
> >
> > I thank the authors for their response. The new cross-domain experiments are indeed interesting and offer more evidence about the benefit of the proposed method. I think it would be useful to include them in the final version. Based on this, I will increase my rating for the paper.

---

### Author Response · Authors · 2021-08-10
**Summary of Reviews**

We thank all reviewers for their time and thoughtful comments on our submission. Here we highlight remarks shared across reviewers.

Overall, all reviewers agreed on the clarity, novelty, and thoroughness of our analysis of episodic sampling in few-shot classification.

Reviewers **3S5L** and **KvDG** asked why we used negative log-likelihood (NLL) over other measures of difficulty, like accuracy. We focused on NLL because (i) it is readily available during training, (ii) it leads to simple yet efficient modeling assumptions, and (iii) it is better suited than accuracy for few-shot episodes. In particular, accuracy artificially discretizes difficulty values because of the low number of query-shots.

Reviewer **KvDG** asked for intuition on why UNIFORM sampling performed best. A uniform distribution over difficulty is akin to putting an uninformative prior of the set of test tasks, which is the safest choice when the test task distribution is unknown. This intuition can be formalized, as done in [1]. We will expand on our explanation in the main text.

Reviewers **wemo** and **sGro** wondered if UNIFORM's improvement over baseline sampling (used in most few-shot works) were significant enough. On average, uniform sampling improves by +2.39% (and +1.85% for the online formulation) which is similar to MetaOptNet's improvement over ProtoNet (+2.35%). Reviewer **sGro** specifically asked why the gains for UNIFORM were smaller in the 1-shot compared to the 5-shot scenarios. We added new analysis showing that the computed training loss is inherently more noisy in the 1-shot scenarios, which reduces the benefits of the IS-based uniform estimator.
Nonetheless, our overall positive results show that episodic sampling matters for few-shot classification, and we hope our work can inspire future research in this direction.

Finally, we are thankful for Reviewer **KvDG**'s suggestion to look into cross-domain transfer. We have added 64 new results when training on Mini-ImageNet and testing on CUB-200, Describable Textures, FGVC-Aircraft, and VGG Flowers. These results further highlight the importance of sampling in episodic training, since UNIFORM (online) outperformed baseline sampling in 49/64 scenarios, was comparable in 12/64 scenarios, and underperformed in only 3/64 scenarios.

**References**

1. Fallah et al., "Generalization of Model-Agnostic Meta-Learning Algorithms: Recurring and Unseen Tasks", arXiv 2021.

---

### Decision · Program_Chairs · 2021-09-27

**Decision:**

Accept (Spotlight)

**Comment:**

The submission investigates the question of how to sample episodes for few-shot learning. By introducing a measure of episode difficulty based on the few-shot learner's NLL on the episode, the authors experiment with importance sampling to simulate different episode difficulty distributions. They observe that without any intervention, episode difficulty is normally distributed, and that a uniform distribution during meta-training yields better results.

Reviewers found the paper well-written and easy to read, and found the proposed approach original and easy to implement. Even though the improvement margins are not substantial, reviewers thought that the ideas and experiments have a broad enough appeal to the few-shot learning community to make the submission a valuable contribution. I therefore recommend acceptance.